# Effect of bilateral contraction on the ability and accuracy of rapid force production at submaximal force level

**Yoichi Ohta** *

Faculty of Social Welfare, Department of Health and Welfare, Shizuoka University of Welfare, Yaizu, Japan

* t_uzura76@live.jp, yohta@suw.ac.jp

## Abstract

The present study aims to clarify the effects of bilateral contraction on the ability and accuracy of rapid force production at the submaximal force level. Eleven right-handed participants performed rapid gripping as fast and precisely as they could in unilateral (UL) and bilateral (BL) contractions in a standing position. Participants were required to impinge a grip force of 30% and 50% of their maximal voluntary contraction (MVC). Ability and accuracy of rapid force production were evaluated using the rate of force development (RFD) and force error, respectively. The data analysis did not observe a significant difference in the RFD between UL and BL contractions in both 30% (420±86 vs. 413±106%MVC/s, p = 0.34) and 50% of MVC (622±84 vs. 619±103%MVC/s, p = 0.77). Although the RFD to peak force ratio (RFD/PF) in BL contraction was lower than in UL in 30% of MVC (12.8±2.8 vs. 13.4±2.7, p = 0.003), it indicated a small effect size (d = 0.22) of the difference between UL and BL in RFD/PF. The absolute force error of BL contraction was higher than of UL contraction in 30% (4.67±2.64 vs. 3.64±1.13%MVC, p = 0.005) and 50% of MVC (5.53±2.94 vs. 3.53±0.71%MVC, p = 0.009). In addition, medium and large effect sizes were observed in absolute force error from 30% (d = 0.51) and 50% of MVC (d = 0.94), respectively. In conclusion, results indicated that the bilateral contraction reduced in the ability and accuracy of rapid force production at the submaximal force level. Nevertheless, the present results suggest that the noticeable effect of bilateral contraction is more prominent on the accuracy than in the ability of rapid force production at the submaximal force level.

## Introduction

The ability of rapid force production is important in several physical activities. In particular, the ability to quickly produce force within a limited range in a short time is required in sport-like situations. For example, a grip force with a latency of less than 100 ms is reported before a ball-racket impact in tennis [1]. The explosive grip force that can be exerted within a 100 ms latency from the onset is approximately 40% of maximal gripping strength [2,3]. Moreover, the production of force intensity immediately before the impact of a cricket bat swing was approximately 36% of their maximum value [4]. Therefore, the explosive force of maximal

**Data Availability Statement:** All relevant data are within the manuscript and its Supporting Information files.

**Funding:** The work was supported JSPS KAKENHI Grant Number 17K13161. The funder had no role

in study design, data collection and analysis, decision to publish, or preparation of the manuscript.

**Competing interests:** The authors have declared that no competing interests exist.

voluntary force production and the ability to produce rapid force at a submaximal force level are important in sport situations. In addition, Li and Turrell [5] suggested that if the force production intensity is inappropriate in racket sports, more energy is used and fatigue will rapidly occur causing the energy transfer to the racket to be poor. Thus, in the rapid force production at a submaximal force level, the ability to exert rapid force and its accuracy intensity are both important.

Previous studies reported that the sum of values of the maximal force of both hands under bilateral contraction produces a low force compared to the sum of values produced under the unilateral maximal force of right- and left-hand (for a review see [6,7]). This phenomenon is referred to as the bilateral deficit (BLD). BLD was observed in maximal force production and the rate of force development (RFD) of the voluntary explosive force production [8–14]. However, previous studies mainly focused on the force development phase during the near maximal or maximal force production (exceed 80% of MVC). There are few studies which investigate the effect of bilateral contraction on rapid force performance at low and middle force levels. Moreover, with regards to the accuracy of force intensity during submaximal rapid force production, a low force intensity error in the bilateral contraction compared to unilateral contractions was reported [15]. Another previous study reported non-difference between bilateral and unilateral tasks [16]. However, no systematic tests were performed to evaluate in detail the accuracy of the bilateral rapid force production at the submaximal force level because the two previous studies did not consider the maximum muscle strength decrease by the BL contraction. Thereby, the target force of the bilateral task is based on the maximum force of the unilateral conditions in the previous studies [15,16]. Therefore, the effect of bilateral contraction on rapid force production ability and accuracy at the submaximal force level remain unclear.

In the maximal handgrip contraction, no bilateral deficit was observed for rapid force [14]. Thus, in the submaximal force levels, the present study hypothesized that bilateral deficit is not observed for rapid force production ability. In addition, right- and left-hand forces in BL contraction showed parallel changes during the entire series of repetitions [15]. Thus, the ability of rapid force production of both hands will be parallel to changes in BL contraction.

This study aims to clarify the effect of bilateral contraction on the ability and accuracy of rapid force production at the submaximal force level.

## Methods

### Participants

Eleven right-handed healthy participants (2 males and 9 females) participated in this study (age: 21.7 ± 2.3 years old; height: 163.6 ± 7.2 cm; weight: 59.3 ± 6.1 kg). The handedness was determined by the writing hand. The present study proceeded in accordance with the Declaration of Helsinki and was approved by the Ethics Committee of Shizuoka University of Welfare, Japan. All participants received a full explanation of the objectives of the investigation and were informed of the experimental procedures in advance. The participants provided written consent to participate in the study.

### Apparatus for grip force measurement

The digital handgrip strength dynamometers (T.K.K.5710b model; Takei Scientific Instruments Co. Ltd., Niigata, Japan) were used to measure the right and left hands grip forces. These dynamometers were connected to strain amplifiers (T.K.K.1268 model; Takei Scientific Instruments Co. Ltd., Niigata, Japan). The measuring range of the dynamometers and strain gauge is 0–100 kg, and the measurement error is ± 0.2% FS or less. The strain amplifiers were

connected to a Power-Lab 16sp analog to digital converter (AD Instruments Pty. Ltd., Bella Vista, Australia), and the grip force signals were recorded with a laptop computer using Chart8.1.13 software via a Power-Lab with a 1-kHz sampling rate. The force signals were submitted to a low-pass filter with a cut-off frequency of 50 Hz, and the data was analyzed using Chart8.1.13 software. The force signals of right- and left-hand were displayed on a 46-inch digital screen (LCD-V463-N2; NEC, Tokyo, Japan) in front of the participant.

### Experimental procedures

Participants visited the laboratory twice, once for a familiarization session and then for the experimental session with a minimum of 24 h between visits. During the familiarization session, the participants held a handgrip strength dynamometer with the right and/or left hand while standing. They practiced the rapid force production by gripping a dynamometer in unilateral (UL) contraction of each right- and left-hand and bilateral (BL) contraction. The participants were instructed not to touch the arm on the body when producing force. They were instructed that force production strategy was to perform force production as fast as possible and to perform rapid relaxation. During this practice, the produced force signal was monitored on a digital screen, which was seen by the participant. The screen was placed 1.5 m in front of the participant (Fig 1). After the rapid force production practice, the grip strength for maximal voluntary contraction (MVC) was determined by each participant grabbing the dynamometer at maximum strength twice. The higher value was considered the MVC. In both UL and BL contractions, MVC was determined by the right- and left-hand, respectively. The MVC trials were performed in random order, with at least 3 min rest between trials.

Participants were shown a line on a digital screen with information from the dynamometer. The screen showed a target line that represented the required grip strength (30% or 50% of MVC). Participants were required to practice so their peak grip strength (height force level in the force-time curve) was 30% or 50% of the MVC, indicated by the target line. For this task, they were asked to generate a force "as fast and precisely as possible" (Fig 2). The force production intensities referred to previous studies [2–4,17]. The onset of the force exertion was arbitrary. In the UL contraction, the target level of 30% and 50% of MVC of right- and left-hand were determined by MVC of right- and left-hand at UL contraction, respectively. In the BL contraction, the target level of 30% and 50% of MVC of right- and left-hand were determined by MVC of right- and left-hand at BL contraction, respectively. Participants were asked to generate the force as fast and precisely as possible with both hands at the same time in the BL contraction task. This fast and precise force exertion task of UL and BL contractions were used in the experimental session.

The participant returned to the laboratory for the experimental session at least 24 h after the familiarization session. First, participants practiced the rapid force production of the submaximal force level with UL and BL contractions in front of a digital screen while standing. After 5 min of rest, the MVC of each participant was determined in the same manner as in the familiarization session. Subsequently, they repeated the task of fast and precisely force exertion of UL and BL contractions with a 30% and 50% target force from the MVC. This task followed the same instructions and manner for the familiarization session. After the fast and precise gripping task practice, they rested for 5 min.

In the main experimental session, participants performed the fast and precisely gripping task in 10 trials either of UL right-hand ($UL_R$), UL left-hand ($UL_L$), or BL contractions. The onset of the force exertion was arbitrary and it was at least 3 s apart. 10 trials were repeated in nine sets (90 trials in total), and $UL_R$, $UL_L$, and BL contractions were included in 3 sets each (30 trials in $UL_R$, $UL_L$, and BL, respectively). $UL_R$, $UL_L$, and BL contractions were performed

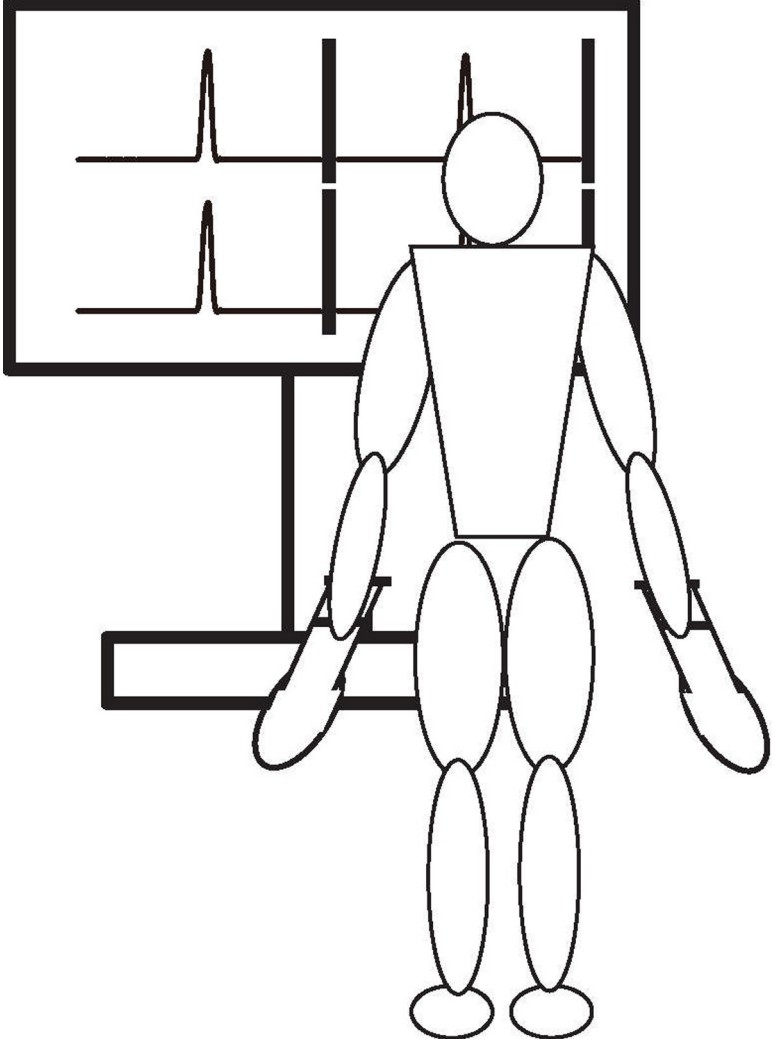

**Fig 1. Experimental condition of the force exerts task of the bilateral contraction.**

in random order, with at least 3 min rest between sets. Three-time rapid contractions before the onset of set were performed to confirm a gripping of the dynamometer. These trials were performed in the target force of both 30% and 50% of MVC in random order, with at least 5 min rest between target forces. The percentage MVC values of right- and left-hand in UL contraction referred to MVC by UL test, and that in BL contraction referred to MVC by BL test, thus the absolute target force values were different between UL and BL contractions. The feedback of force intensity error was shown on a digital screen, although the force error value was not displayed. The participant could see the force curve shape, although the value of the RFD was not displayed. The force curve shape was displayed after the force production. During the experimental task, the force was monitored not to exceed 3% of MVC before rapid force production with the aid of the Chart8.1.13software. Otherwise, the test was redone.

## Detection and analysis of experimental signals

The Bilateral Index (%) of MVC and RFD/PF = ((bilateral right value + bilateral left value)/ (unilateral right value + unilateral left value)) * 100 were determined. The peak force (PF) of

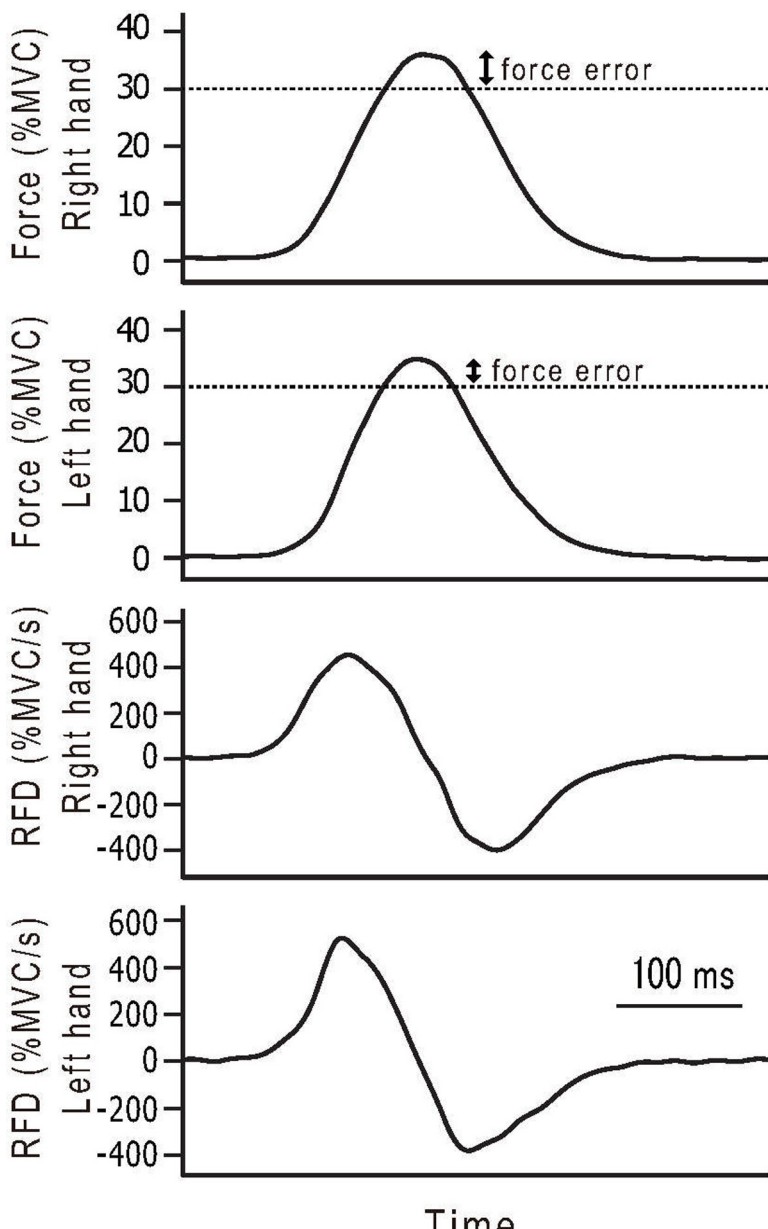

**Fig 2. Example of the time course of force and rate of force development (RFD) measurements in bilateral contraction.** Arrows (↔) show force error.

gripping was chosen as the highest force produced on the trials. The PF was normalized to the MVC of gripping and is expressed as a percentage of the MVC (%MVC). The peak RFD was calculated from the first derivatives of the force signals with a 32-point window width using chart 8.1.13. The RFD was shown as a percentage of MVC per second (%MVC/s). The normalized RFD was calculated by the RFD to PF ratio (RFD/PF) in each trial. In the PF, RFD, and RFD/PF, the mean of the 30 trials was used as the representative value for $UL_R$, $UL_L$, BL right-hand ($BL_R$), and BL left-hand ($BL_L$). The variability of RFD and RFD/PF was measured from the standard deviation for each participant across the 30 trials. Force errors were determined as the difference between the target force (30% and 50% of MVC) and PF in each trial (Fig 2).

Three types of force errors were calculated: constant force error (CFE), absolute force error (AFE), and variable force error (VFE). When PF was greater or lower than the target force, the CFE showed positive or negative values, respectively. The mean value from the 30 trials was used as the representative value for $UL_R$, $UL_L$, $BL_R$, and $BL_L$ in force intensity errors. The equations for CFE, AFE and VFE are as follows [17]:

$$CFE = \left[ \sum (X_i - F)/N \right] \tag{1}$$

$$AFE = \left[ \sum (|X_i - F|)/N \right] \tag{2}$$

$$VFE = \sqrt{\left[ \sum (X_i - M)^2 / N \right]} \tag{3}$$

where $i$ indicates the trial number; $X_i$ is the produced peak force for the $i$th trial; F is the target force level (30% or 50% of MVC); $N$ is the number of trials, and $M$ is the average peak force for the $N$ trials.

## Statistical analysis

All data are presented as the mean value with the standard deviation (SD). A one-sample t-test against 100 was used to determine if the Bilateral Index was significantly different from 100. The effects of the bilateral contraction (unilateral vs. bilateral contraction) and of hands (right vs. left hand) were determined by employing the two-way repeated-measures analysis of variance (ANOVA). A simple main effect test was applied if the interaction effect was significant. The Pearson's product-moment correlation coefficient (r) was used to determine correlations between right- and left-hand during the entire repetitions of 30 trials in both UL and BL contractions in inter-individual data (n = 30) in PF, RFD, and PFD/PF. The differences of correlation coefficient among parameters (PF vs. RFD vs. PFD/PF) and contraction intensity (30% vs. 50%) in BU contraction was determined by applying the two-way repeated-measures analysis of variance (ANOVA). The Statistical Package for the Social Sciences (SPSS for Windows, version 22.0, IBM, USA) was used for all statistical analyses. Data including effect size ($\eta_p^2$) was statistically analyzed using SPSS for Windows. The effect size (d) was calculated using Cohen's d index [18,19].

# Results

## Maximal gripping force

Maximal gripping force of the $UL_R$, $UL_L$, $BL_R$, and $BL_L$ were 32.1 ± 9.0, 29.0 ± 7.5, 30.9 ± 8.5, and 28.3 ± 9.8 kg, respectively. The Bilateral Index (96.3 ± 5.4%) was significantly lower than 100% (p = 0.044). The effect size of the difference between the sum of the $UL_R$ and $UL_L$ of maximal gripping force and sum of $BL_R$, and $BL_L$ using Cohen's d was 0.11, which is a trivial size.

## RFD and RFD/PF

Fig 3A–3D shows the means ± SD of the RFD and variability of the RFD in $UL_R$, $UL_L$, $BL_R$, and $BL_L$ in 30% and 50% of the MVC. The 2-way ANOVA of the RFD of the 30% of MVC (Fig 3A) showed no significant main effects of bilateral contraction [F (1, 10) = 0.97, p = 0.34, $\eta_p^2$ = 0.08] and hands [F (1, 10) = 0.71, p = 0.41, $\eta_p^2$ = 0.06], and no interactions [F (1, 10) = 0.37, p = 0.55, $\eta_p^2$ = 0.03]. In the RFD of the 50% of MVC (Fig 3C), there were no significant bilateral contraction main effects [F (1, 10) = 0.08, p = 0.77, $\eta_p^2$ = 0.009] and hands [F (1, 10) =

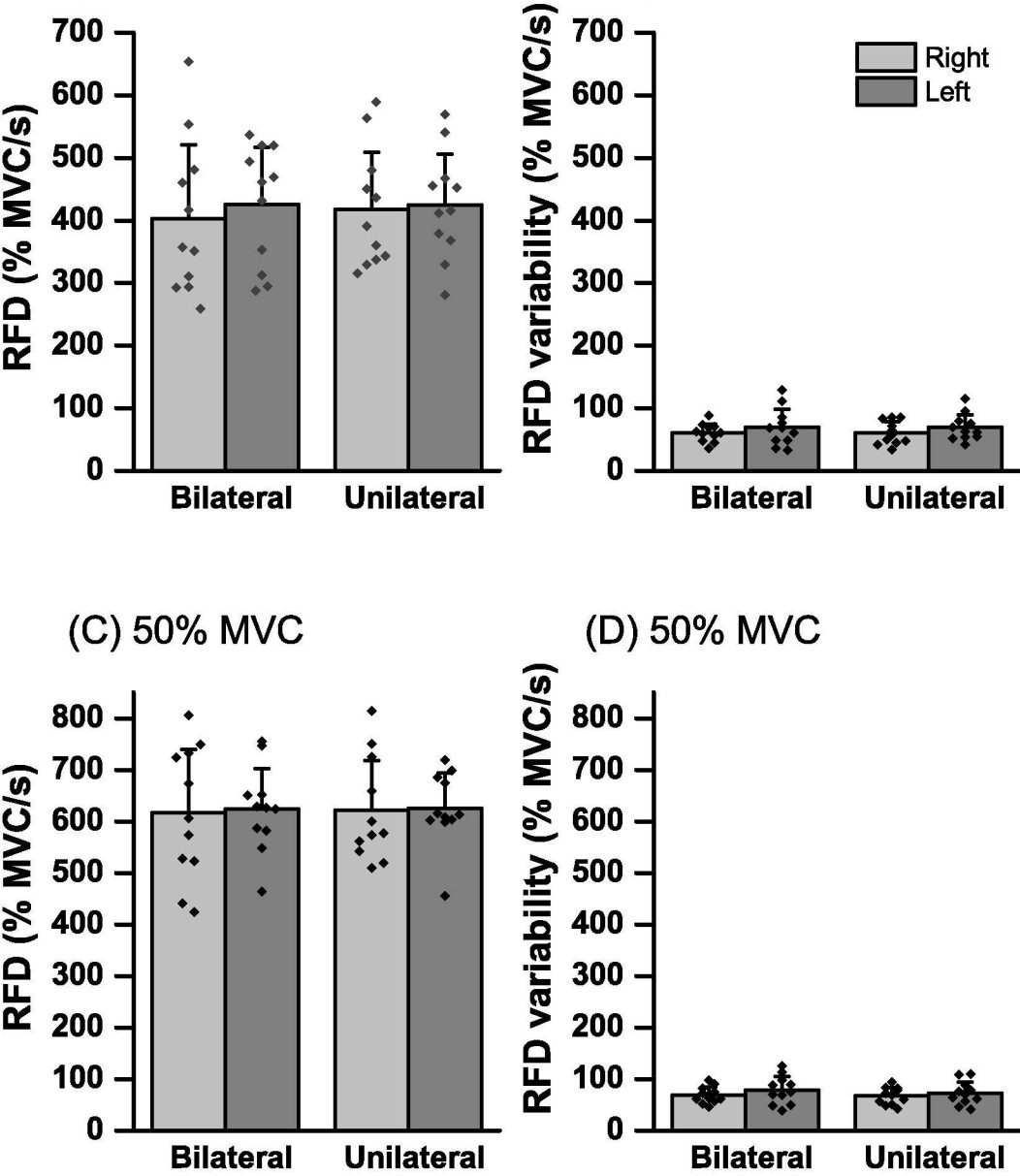

**Fig 3. Mean rate of force development (RFD) in right- (gray bar) and left-hand (dark gray bar) for bilateral and unilateral contractions.** Dot presents the value for each individual. (**A**) mean peak RFD of 30% of maximal voluntary contraction (MVC) target, (**B**) RFD variability of 30% of MVC target, (**C**) mean peak RFD of 50% of MVC target and (**D**) RFD variability of 50% of MVC target.

0.06, p = 0.79, $\eta_p^2$ = 0.007], and no interactions [F (1, 10) = 0.017, p = 0.90, $\eta_p^2$ = 0.002]. Regarding the variability of RFD of the 30% of MVC (Fig 3B), there were no significant main bilateral contraction effects [F (1, 10) = 0.005, p = 0.94, $\eta_p^2 < 0.001$] and hands [F (1, 10) = 3.44, p = 0.09, $\eta_p^2$ = 0.007], and no interactions [F (1, 10) < 0.001, p = 0.99, $\eta_p^2 < 0.001$]. In the RFD variability of the 50% MVC (Fig 3D), there were no significant main bilateral contraction effects [F (1, 10) = 0.87, p = 0.37, $\eta_p^2$ = 0.08] and hands [F (1, 10) = 1.29, p = 0.28, $\eta_p^2$ = 0.11], and no interactions [F (1, 10) = 0.22, p = 0.64, $\eta_p^2$ = 0.02].

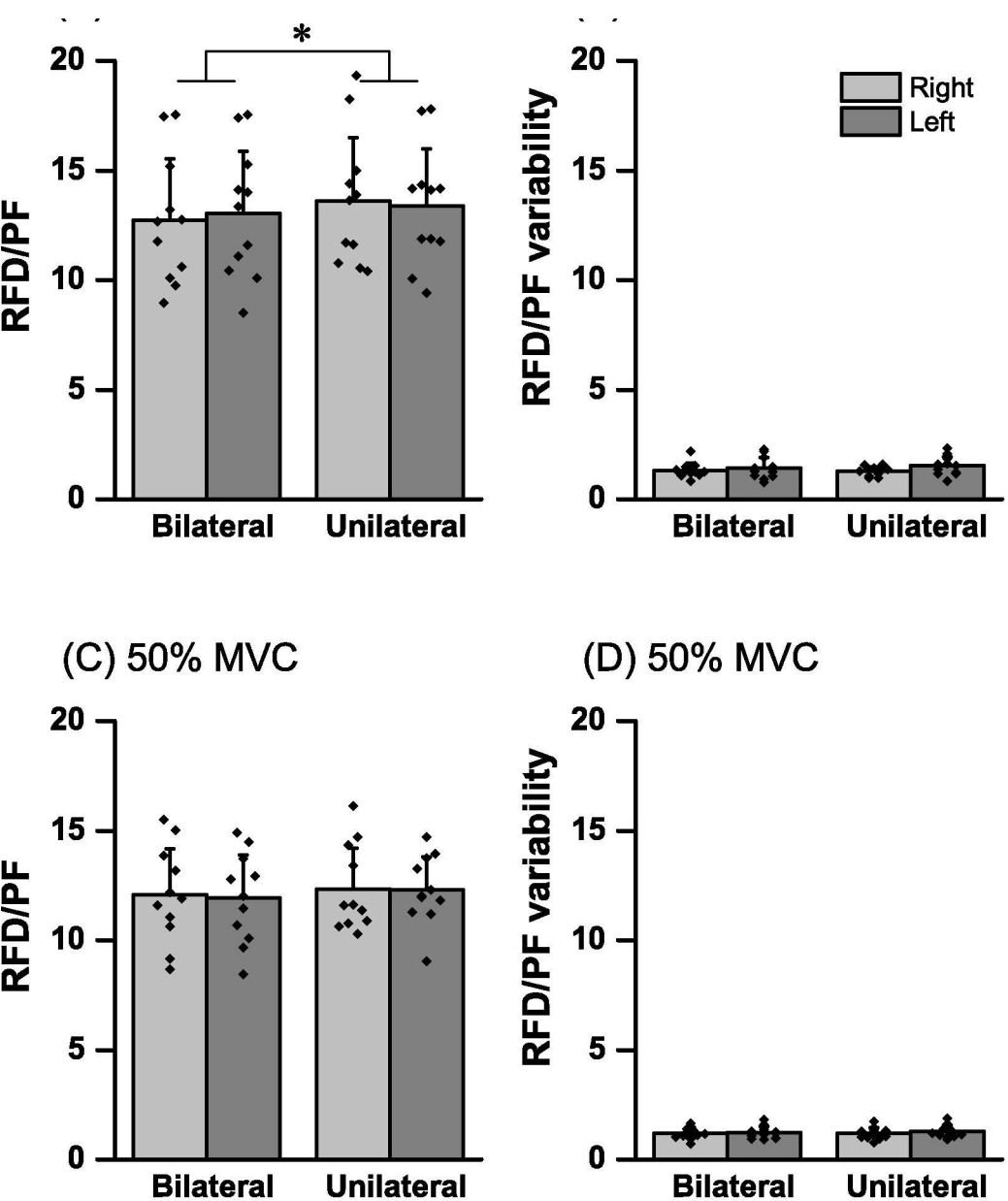

**Fig 4. Mean rate of force development (RFD) to peak force ratio (RFD/PF) in right- (gray bar) and left-hand (dark gray bar) for bilateral and unilateral contractions.** Dot presents the value for each individual. (**A**) mean RFD/PF of 30% of maximal voluntary contraction (MVC) target, (**B**) RFD/PF variability of 30% of MVC target, (**C**) mean RFD/PF of 50% of MVC target and (**D**) RFD/PF variability of 50% of MVC target. * show main effect (p < 0.05).

Fig 4A–4D) shows the means ± SD of the RFD/PF and variability of the RFD/PF in $UL_R$, $UL_L$, $BL_R$, and $BL_L$ in 30% and 50% of the MVC. A significant of bilateral contraction main effect in RFD/PF of the 30% of MVC was observed [F (1, 10) = 15.8, p = 0.003, $\eta_p^2$ = 0.61], although no main effect of hands [F (1, 10) = 0.47, p = 0.83, $\eta_p^2$ = 0.005] and no interaction [F (1, 10) = 3.27, p = 0.10, $\eta_p^2$ = 0.24] was observed (Fig 4A). The Bilateral Index (95.22 ± 4.22%) of RFD/PF of 30% of MVC was significantly lower than 100% (p = 0.005). The effect size of difference between sum of the $UL_R$ and $UL_L$ of RFD/PF of 30% of MVC and that of $BL_R$ and $BL_L$ using Cohen's d was 0.22, that is small level. In the RFD/PF of the 50% of MVC (Fig 4C), there

were no significant bilateral contraction main effects [F (1, 10) = 3.82, p = 0.07, $\eta_p^2$ = 0.27] and hands [F (1, 10) = 0.12, p = 0.73, ηp2 = 0.01], and no interactions [F (1, 10) = 0.06, p = 0.80, $\eta_p^2$ = 0.007]. Bilateral Index (96.96 ± 4.83%) for the RFD/PF of 50% of MVC did not differ from 100% (p = 0.075). The effect size of difference between sum of the $UL_R$ and $UL_L$ of RFD/PF of 30% of MVC and that of $BL_R$ and $BL_L$ using Cohen's d index was 0.11, that is small level. Regarding the variability of RFD/PF of the 30% of MVC (Fig 4B), there were no significant bilateral contraction main effects [F (1, 10) = 0.22, p = 0.64, $\eta_p^2$ = 0.02] and hands [F (1, 10) = 2.14, p = 0.17, $\eta_p^2$ = 0.17], and no interactions [F (1, 10) = 2.50, p = 0.14, $\eta_p^2$ = 0.20]. In the RFD/PF variability of the 50% of MVC (Fig 4D), there were no significant bilateral contraction main effects [F (1, 10) = 0.10, p = 0.75, $\eta_p^2$ = 0.01] and hands [F (1, 10) = 0.67, p = 0.42, $\eta_p^2$ = 0.06], and no interactions [F (1, 10) = 0.15, p = 0.70, $\eta_p^2$ = 0.01].

## Force intensity error

Fig 5A–5F shows the means ± SD of the CFE, AFE, and VFE in $UL_R$, $UL_L$, $BL_R$, and $BL_L$ in 30% and 50% of the MVC. The 2-way ANOVA of the CFE of the 30% of MVC showed no significant bilateral contraction main effects [F (1, 10) = 2.30, p = 0.18, $\eta_p^2$ = 0.18] and hands [F (1, 10) = 1.35, p = 0.27, $\eta_p^2$ = 0.12], and no interactions [F (1, 10) = 0.08, p = 0.78, $\eta_p^2$ = 0.008] (Fig 5A). In the CFE of the 50% of MVC (Fig 5D), there are no significant bilateral contraction main effects [F (1, 10) = 4.75, p = 0.054, $\eta_p^2$ = 0.32] and hands [F (1, 10) = 0.62, p = 0.44, $\eta_p^2$ = 0.05], and no interaction significant interactions [F (1, 10) = 0.26, p = 0.61, $\eta_p^2$ = 0.02].

Regarding the AFE of the 30% of MVC (Fig 5B), the bilateral contraction main effect [F (1, 10) = 12.50, p = 0.005, $\eta_p^2$ = 0.56] was significant although there were no main effect of hands [F (1, 10) = 1.90, p = 0.20, $\eta_p^2$ = 0.16] and no interactions [F (1, 10) = 0.32, p = 0.58, $\eta_p^2$ = 0.03]. The effect size of difference between sum of the $UL_R$ and $UL_L$ of AFE of 30% of MVC and that of $BL_R$ and $BL_L$ using Cohen's d index was 0.51, which is considered as a medium level. In the AFE of the 50% of MVC (Fig 5E), the bilateral contraction main effect [F (1, 10) = 10.53, p = 0.009, $\eta_p^2$ = 0.51] was significant although there was no main effect of hands [F (1, 10) = 0.41, p = 0.53, $\eta_p^2$ = 0.04], and no interactions [F (1, 10) = 0.78, p = 0.39, $\eta_p^2$ = 0.07]. The effect size of difference between sum of the $UL_R$ and $UL_L$ of AFE of 50% of MVC and that of $BL_R$ and $BL_L$ using Cohen's d index was 0.94, what is considered as a large level.

Regarding the VFE of the 30% of MVC (Fig 5C), there were no significant bilateral contraction main effects [F (1, 10) = 0.42, p = 0.53, $\eta_p^2$ = 0.04] and hands [F (1, 10) = 3.25, p = 0.10, $\eta_p^2$ = 0.24], and no interactions [F (1, 10) = 0.53, p = 0.48, $\eta_p^2$ = 0.05]. In VFE of the 50% of MVC (Fig 5F), there was no significant bilateral contraction main effects [F (1, 10) = 1.50, p = 0.24, $\eta_p^2$ = 0.13] and hands [F (1, 10) = 0.53, p = 0.48, $\eta_p^2$ = 0.05], and no interactions [F (1, 10) = 3.28, p = 0.10, $\eta_p^2$ = 0.24].

## Right- and left-hand correlations in PF, RFD and RFD/PF

Table 1 shows average of correlation coefficients between right- and left-hand during the entire repetitions of 30 trials in all participants in the PF, RFD and RFD/PF of the UL and BL contractions. Significant levels of correlation coefficients were observed in PF, RFD and RFD/PF in the BL contraction in 30% and 50% of MVC, although not in the UL contraction. The 2-way ANOVA of the correlation coefficients of parameters (PF, RFD, and RFD/PF) in both contraction intensity (30% and 50% of MVC) in BL contraction showed no significant main effects of parameters [F (1, 10) = 1.42, p = 0.26, $\eta_p^2$ = 0.12] and contraction intensity [F (1, 10) = 0.16, p = 0.69, $\eta_p^2$ = 0.01], and no interactions [F (1, 10) = 0.93, p = 0.40, $\eta_p^2$ = 0.08].

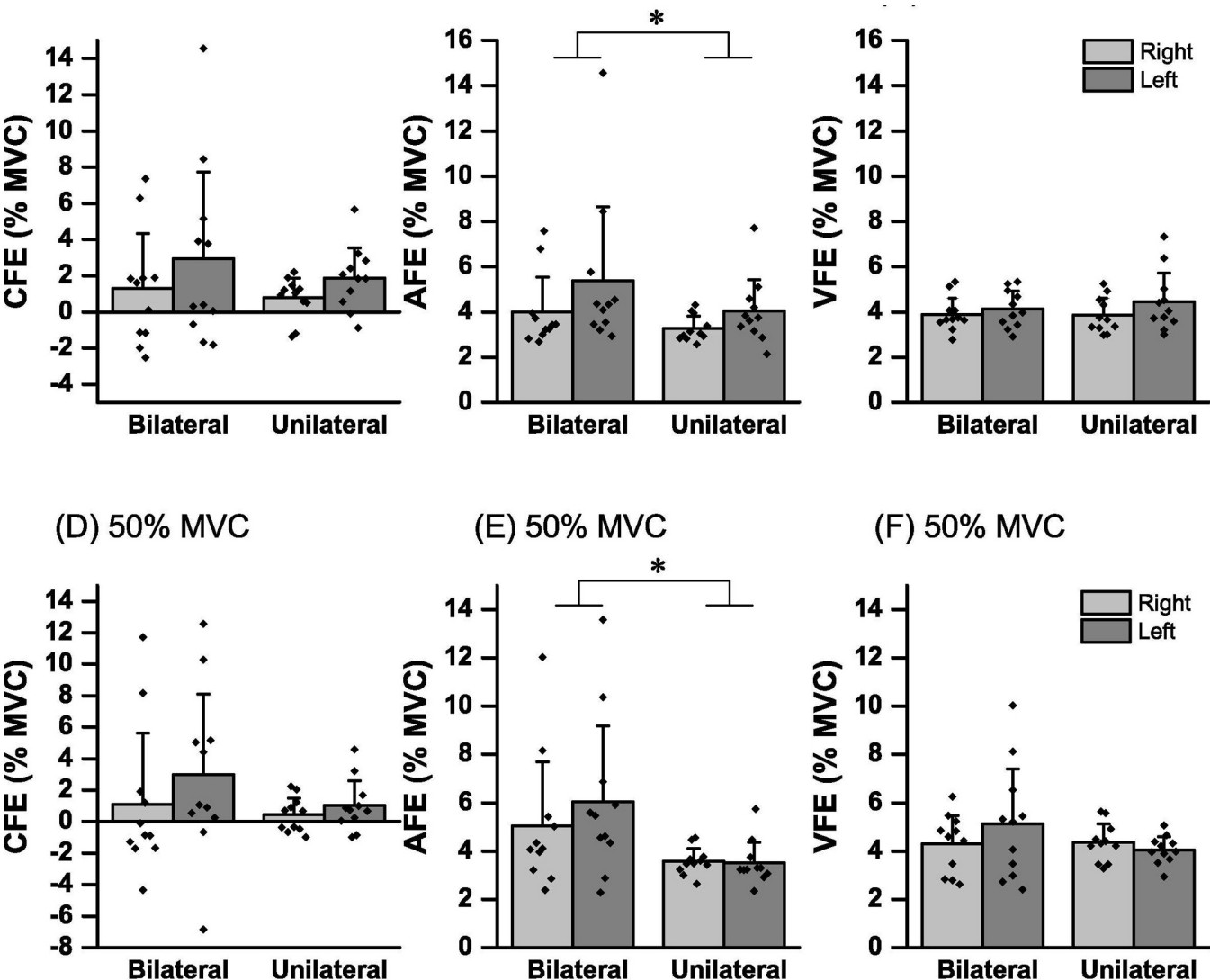

**Fig 5. Mean force intensity errors in right- (gray bar) and left-hand (dark gray bar) for bilateral and unilateral contractions.** Dot presents the value for each individual. (**A**) constant force error (CFE) of 30% of maximal voluntary contraction (MVC) target, (**B**) absolute force error (AFE) of 30% of MVC target, (**C**) variable force error (VFE) of 30% of MVC target, (**D**) CFE of 50% of MVC target, (**E**) AFE of 50% of MVC target, and (**F**) VFE of 50% of MVC target. * show main effect (p < 0.05).

## Discussion

### Effect of bilateral contraction on the ability of rapid and maximal force production

In the present study, wherein participants were instructed to grab a dynamometer with 30% and 50% of MVC as fast and precisely as possible, there was no significant difference in peak RFD between UL and BL contractions in both cases of MVC (Fig 3A and 3C). However, these results did not indicate that the ability of rapid force production was not affected by BL contraction because peak RFD is strongly related to produced PF during rapid force production [20]. To eliminate the influence of produced PF, the ability of rapid force production at the submaximal force level is evaluated by the RFD/PF value, which is used as an index of rapid force performance of independent on produced PF [17]. The present study observed that

**Table 1. Average value of correlation coefficients (r) between right- and left-hand during the repetitions of 30 trials in all participants.**

|  | 30% MVC | 50% MVC |
|---|---|---|
| PF of UL (r) | 0.07 ± 0.26 | 0.07 ± 0.19 |
| PF of BL (r) | 0.54 ± 0.17* | 0.47 ± 0.17* |
| RFD of UL (r) | 0.05 ± 0.17 | 0.0004 ± 0.22 |
| RFD of BL (r) | 0.50 ± 0.17* | 0.45 ± 0.17* |
| RFD/PF of UL (r) | 0.01 ± 0.19 | -0.01 ± 0.24 |
| RFD/PF of BL (r) | 0.53 ± 0.15* | 0.57 ± 0.14* |

PF, peak force; UL, unilateral; BL, bilateral; RFD, rate of force development; RFD/PF, RFD to peak force ratio. Values are means ± SD.

*Significant level (r > 0.361, $p < 0.05$, n = 30).

RFD/PF in BL contraction was significantly lower than that in UL contraction in 30% of MVC (Fig 4A), although no significant difference was observed in 50% of MVC (Fig 4C). This is the first study to investigate a potential BLD in the ability of rapid force production at the submaximal force level. These results indicate that the ability of rapid force production at the submaximal force level is reduced by BL contraction at lower contraction intensity. However, the BLD of RFD/PF of 30% of MVC (4.77%) and that of MVC of handgrip strength (3.7%) in the present study were relatively lower than previously studies [8,12,21,22]. In addition, in both RFD/PF and MVC, the effect size of difference between UL and BL indicated a small level (d < 0.22). These results suggest that the effect of BL contraction on the abilities of rapid and maximal force production of this study were low.

In the previous study of the handgrip strength, BLD has been noticeably observed in the state where the forearm was fixed in the sitting position [8,12,21,22], although it has been reported that BLD is not observed in standing and supine positions [23–25]. The BLD was explained by several psychological and physiological mechanisms in previous studies (for a review see [6,7]). Based on another point of view, the previous studies suggested that the BLD in maximal strength was a strong influence of postural stability and mechanical configuration of the dynamometer [11,26]. Simoneau-Buessinger, Leteneur [26] revealed that the BLD in maximal strength is due to the additional torque from the body adjustments. The participant was instructed not to touch the arm to the body when doing the experiments in the standing position. Thus, the conditions imply less addition of the torque from body adjustments than the hand griping in a state of arm fixed in sitting position. Differences in such measurement conditions may contribute to the small effect of BL contraction on the ability of rapid and maximal force production. However, in the standing position, BLD has been shown in elderly subjects and the right hand only in the young subjects [27]. There are few studies on the effect of handgrip strength performance on BL contraction in standing position compared to that of sitting position. In sports and daily life, since the muscle force output is usually performed in standing position rather than fixed sitting position, further research is required to fully understand the effects of BL contraction on the ability of rapid force production including factors such as gender and age in sports and daily life situations.

## Effect of bilateral contraction on intensity accuracy of rapid force production

The previous study reported that the accuracy of submaximal rapid force was lager during BL contraction than during UL for the left hand only [15]. Conversely, other previous study

reported no difference between bilateral and unilateral tasks [16]. In the present study, no significant effect of BL contraction on CFE and VFE was observed, although AFE of BL contraction was significantly greater than UL contraction in 30% and 50% of MVC (Fig 5A and 5D). Moreover, the present study observed medium and large effect sizes in AFE of 30% and 50% of MVC, respectively. The CFE, AFE and VFE were used as an index of the trend, accuracy and intrapersonal variability of rapid force production intensity, respectively. Thus, the present result suggests that the effect of BL contraction on trend and intrapersonal variability of rapid force production intensity is low, although the accuracy of rapid force production intensity decreases with BL contraction. The cause of the different results between previous [15,16] and present studies may be due to differences of the setting method of target force in the bilateral task, muscle and posture. Especially, the difference of the setting method of target force in the bilateral task is considered to have a significant impact on the results of the accuracy. In the previous studies, the target force of the bilateral task is based on the maximum force of the unilateral conditions [15,16]. In this condition, the comparison of accuracy in BL and UL contraction is not sufficient. In particular, when the tendency to exert too much force than the target is strong such as previous study [15], the possibility of exerting muscle strength close to the target by the decrease of muscle strength due to BL contraction is undeniable. Therefore, the result of the accuracy of this study may be more accurate capturing the effect of BL contraction on the accuracy of force intensity during submaximal rapid force production.

### Effects on intrapersonal variability of rapid force production of bilateral contraction

The results of the variability of this study (VFE, the variability of RFD, and RFD/PF) are intended to show the deviation in 30 trials within the subject, the lower this value, so that it was performed the same force output and/or rapid force production in 30 trials. In the present study, there were no significant differences between UL and BL contractions in VFE (Fig 5C and 5F) and variability of rapid force production parameters (variability of RFD and RFD/PF) in the 30% and 50% of MVC (Figs 3B, 3D and 4B and 4D). Since AFE and RFD/PF were decreased with BL contraction, the results of variability suggest that BL contraction affects the accuracy and ability of rapid force production, although not significantly affect the intrapersonal variability of rapid force production. However, there are few studies that investigate intrapersonal variability in several trials of rapid force production in BL contraction. Thus, the present study proposed that clarifying the effect of bilateral contraction on intrapersonal variability of rapid force parameters with submaximal contraction level will enhance our understanding of the motor control functions of voluntary rapid force production.

### The correlation between right- and left-hand in rapid force production

In agreement with the results from Yamaguchi et al. [16], the present results show that the no significant difference between right- and left-hand in effect of BL contraction on the accuracy of rapid force production. Moreover, the present study found no significant difference between right- and left-hand in effect of BL contraction on the ability of rapid force production parameters (RFD and RFD/PF). In contrast, the previous study found that the force output intensity by both hands during BL contraction showed parallel changes during the entire series of repetitions but not showed in UL contraction [15]. In the present study, a significant correlation was obtained during the entire series of 30 trials between peak forces of right- and left-hand in BL contraction in 30% and 50% of MVC, while the two peak forces were not correlated in the UL contraction (Table 1). In addition, the ability of rapid force production parameters (RFD and RFD/PF) also examined the relationship between right- and left-hand, and found a significant

correlation during the entire series of 30 trials in BL contraction in 30% and 50% of MVC but not UL contraction (Table 1). The previous study suggested that the correlation in force output intensity between right- and left-hand may be due to a common neural drive for submaximal target contractions during BL contraction [15]. The results of the present study support this suggestion, and suggest that a common neural drive for the intensity of rapid force also exists for the ability of rapid force production at submaximal force level in BL contraction.

## Limitations of the study

There are some limitations to this study. Firstly, the sample size is small. The number of participants (11) would not be sufficient statistical power. Moreover, in the previous study examining sex differences in the bilateral deficit, there are reports where sex differences were observed [13] and some were not [14]. Thus, influence of sex on the bilateral index remain unclear. Since the present study is the result of mixed sexes with a large number of females, it will be necessary to consider sex differences in detail in the future. Moreover, the present study is an investigation of only force data. Therefore, to clarifying the effect of bilateral contraction on co-regulating rapid force production ability and accuracy during submaximal contraction, it would be useful to examine using methods that evaluate other neuromuscular functions (e.g. electromyography and mechanomyography).

## Conclusion

The present study suggests that, in the situation where the rapid gripping in standing position, bilateral contraction is reduced in both the ability and accuracy of rapid force production during submaximal force level. However, the small effect size in the ability of rapid force production and observed medium and large effect sizes in the accuracy of rapid force production intensity have been observed. Thus, the present results suggest that the noticeable effect of bilateral contraction is observed on accuracy than the ability of rapid force production at a submaximal force level.

## Supporting information

**S1 Dataset.**
(XLSX)

## Author Contributions

**Conceptualization:** Yoichi Ohta.

**Data curation:** Yoichi Ohta.

**Formal analysis:** Yoichi Ohta.

**Funding acquisition:** Yoichi Ohta.

**Investigation:** Yoichi Ohta.

**Methodology:** Yoichi Ohta.

**Project administration:** Yoichi Ohta.

**Resources:** Yoichi Ohta.

**Validation:** Yoichi Ohta.

**Visualization:** Yoichi Ohta.

**Writing – original draft:** Yoichi Ohta.

**Writing – review & editing:** Yoichi Ohta.

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
