## [Decision Letter · Decision Letter 0]

3 Nov 2020

PONE-D-19-35654

Effect of bilateral contraction on the ability and accuracy of rapid force production at submaximal force level

PLOS ONE

Dear Dr. Ohta,

Thank you for submitting your manuscript to PLOS ONE. After careful consideration, we feel that it has merit but does not fully meet PLOS ONE’s publication criteria as it currently stands. Therefore, we invite you to submit a revised version of the manuscript that addresses the points raised during the review process.

We look forward to receiving your revised manuscript.

Kind regards,

Riccardo Di Giminiani

Academic Editor

PLOS ONE

Journal Requirements:

Reviewers' comments:

Reviewer's Responses to Questions

**Comments to the Author**

1. Is the manuscript technically sound, and do the data support the conclusions?

Reviewer #1: Yes

Reviewer #2: Yes

2. Has the statistical analysis been performed appropriately and rigorously? 

Reviewer #1: Yes

Reviewer #2: Yes

3. Have the authors made all data underlying the findings in their manuscript fully available?

Reviewer #1: Yes

Reviewer #2: Yes

4. Is the manuscript presented in an intelligible fashion and written in standard English?

Reviewer #1: Yes

Reviewer #2: Yes

5. Review Comments to the Author

Reviewer #1: Well designed and well presented manuscript. The authors successfully achieved their stated purpose. The authors should replace the bar plots in figures 3-5 with raincloud plots. Below are some links on raincloud plots:

https://wellcomeopenresearch.org/articles/4-63#:~:text=In%20essence%2C%20raincloud%20plots%20combine,error%2C%20such%20as%20a%20boxplot.

https://micahallen.org/2018/03/15/introducing-raincloud-plots/

https://github.com/RainCloudPlots/RainCloudPlots

Reviewer #2: Comments for “Effect of bilateral contraction on the ability and accuracy of rapid force production at submaximal force level”

The purpose of the study was to examine the effects of bilateral contraction on the ability and accuracy of rapid force development (RFD) at the submaximal force level. Eleven subjects performed unilateral (UL) and bilateral (BL) grip contractions at the 30% and 50% of the MVC. No differences were found for RFD between UL and BL contractions in both 30% and 50% MVC. The absolute force error of BL contraction was higher than UL contraction in 30% and 50% of MVC.

One major concern is that the study only examined the force data. If there were other mechanistic variables measured (e.g., rate of EMG rise, etc.), it will be a much-completed nice study. Additionally, one experimental setup regarding the target force (30% and 50% MVC) contraction has flaws and could be influencing the results (see my comments in the later section). Following are my detailed comments:

Abstract

Please add mean and standard deviation values as well as the p-values when reporting the results. In addition, the author should also report the effect size value.

Introduction

Lines 55-57, the author missed a few more recent literatures regarding the BLD in peak force and RFD:

Carr, J. C., Bemben, M. G., Black, C. D., Ye, X., & DeFreitas, J. M. (2020). Bilateral deficit in strength but not rapid force during maximal handgrip contractions. European Journal of Sport Science, 1-8.

Ye, X., Miller, W. M., Jeon, S., & Carr, J. C. (2019). Sex comparisons of the bilateral deficit in proximal and distal upper body limb muscles. Human movement science, 64, 329-337.

Lines 59-60, this statement is not accurate. There are a few recent articles regarding submaximal contractions and accuracy:

Brustio, P. R., Casale, R., Buttacchio, G., Calabrese, M., Bruzzone, M., Rainoldi, A., & Boccia, G. (2019). Relevance of evaluating the rate of torque development in ballistic contractions of submaximal amplitude. Physiological measurement, 40(2), 025002.

Boccia, G., Dardanello, D., Brustio, P. R., Tarperi, C., Festa, L., Zoppirolli, C., ... & Rainoldi, A. (2018). Neuromuscular fatigue does not impair the rate of force development in ballistic contractions of submaximal amplitudes. Frontiers in physiology, 9, 1503.

Lines 70-71, only using the force data would still be difficult to have a better understanding of BLD.

Lines 72-73, what made the author have such hypothesis? Please comment and clarify.

Methods

Participants: how was the sample size (11) determined?

It was also reported that females and males may have the different bilateral index, especially for the hand muscle (see study: Ye, X., Miller, W. M., Jeon, S., & Carr, J. C. (2019). Sex comparisons of the bilateral deficit in proximal and distal upper body limb muscles. Human movement science, 64, 329-337.). Please justify why you used the mixed sexes for your experiment.

Lines 121-124, and lines 155-157, it was suggested that targeting the force levels can slow the rate of force production (Gordon, J., Ghez, C. Trajectory control in targeted force impulses. Exp Brain Res 67, 241–252 (1987). https://doi.org/10.1007/BF00248546). How can the author ensure the RFD in the current study was not influenced by the experimental setup?

Line 165, please explain the “32-point window”, how long is this window?

In addition, when calculating RFD, how was the onset of the force signal determined? I could not find the description.

Line 170, when calculating force error, what was the time window used in the calculations?

Results and Discussion

Overall, the results are hard to follow.

Line 207, Cohen’s d = 0.11, belongs to trivial size.

Please change all the “effect size index” to “effect size”. Also, consider using small, medium, large to describe the effect size.

For the discussion, limitations of the experiment should be pointed out.

6. PLOS authors have the option to publish the peer review history of their article (what does this mean?). If published, this will include your full peer review and any attached files.

Reviewer #1: No

Reviewer #2: No

---

## [Author Response · Author response to Decision Letter 0]

25 Nov 2020

Response of the Reviewers' comments:

 Thank you very much for your consideration and valuable suggestions for improving my manuscript. I have read the reviewers’ comments very carefully, and have made the following changes according to the comments and suggestions of the reviewers. Moreover, I found mistake, so I corrected the notation of the result (Lines 258-261). The revised page numbers and line numbers are indicated following my response to each of the comments. I hope that the revised manuscript satisfactorily addresses the issues raised by the reviewers. My responses are shown in text starting with RE.

Response to Reviewer: 1

Reviewer #1: Well designed and well presented manuscript. The authors successfully achieved their stated purpose. The authors should replace the bar plots in figures 3-5 with raincloud plots. Below are some links on raincloud plots:

https://wellcomeopenresearch.org/articles/4-63#:~:text=In%20essence%2C%20raincloud%20plots%20combine,error%2C%20such%20as%20a%20boxplot.

https://micahallen.org/2018/03/15/introducing-raincloud-plots/

https://github.com/RainCloudPlots/RainCloudPlots

RE: According to the reviewer’s comment, I replaced the bar plots in figures 3-5 with raincloud plots.

Response to Reviewer: 2

Reviewer #2: Comments for “Effect of bilateral contraction on the ability and accuracy of rapid force production at submaximal force level”

The purpose of the study was to examine the effects of bilateral contraction on the ability and accuracy of rapid force development (RFD) at the submaximal force level. Eleven subjects performed unilateral (UL) and bilateral (BL) grip contractions at the 30% and 50% of the MVC. No differences were found for RFD between UL and BL contractions in both 30% and 50% MVC. The absolute force error of BL contraction was higher than UL contraction in 30% and 50% of MVC.

One major concern is that the study only examined the force data. If there were other mechanistic variables measured (e.g., rate of EMG rise, etc.), it will be a much-completed nice study. Additionally, one experimental setup regarding the target force (30% and 50% MVC) contraction has flaws and could be influencing the results (see my comments in the later section). Following are my detailed comments:

RE: Although EMG is a useful tool for examining individual muscle activities, there are many muscles related to handgrip strength. Because it is difficult to measured many muscle activities related to handgrip strength, I targeted muscle strength, which is the final output of various muscle activities. Moreover, the present study examined the ability and accuracy of rapid force production in the situation where the submaximal target force levels (30% and 50% MVC). Because the rapid force production below 50% MVC has been confirmed in racket sports, it is not considered that the experimental setup in this study is a flaw.

Stretch R, Buys F, Viljoen G. The kinetics of the drive off the front foot in cricket batting: hand grip force. South African Journal for Research in Sport, Physical Education and Recreation. 1995;18(2):83-93.

Knudson DV, White SC. Forces on the hand in the tennis forehand drive: application of force sensing resistors. Journal of Applied Biomechanics. 1989;5(3):324-31.

Abstract

Please add mean and standard deviation values as well as the p-values when reporting the results. In addition, the author should also report the effect size value.

RE: According to the reviewer’s comment,　I added mean, standard deviation, p-values, and effect size values in the abstract.

Introduction

Lines 55-57, the author missed a few more recent literatures regarding the BLD in peak force and RFD:

Carr, J. C., Bemben, M. G., Black, C. D., Ye, X., & DeFreitas, J. M. (2020). Bilateral deficit in strength but not rapid force during maximal handgrip contractions. European Journal of Sport Science, 1-8.

Ye, X., Miller, W. M., Jeon, S., & Carr, J. C. (2019). Sex comparisons of the bilateral deficit in proximal and distal upper body limb muscles. Human movement science, 64, 329-337.

RE: According to the reviewer’s comment,　I added literatures (Line 60).

Lines 59-60, this statement is not accurate. There are a few recent articles regarding submaximal contractions and accuracy:

Brustio, P. R., Casale, R., Buttacchio, G., Calabrese, M., Bruzzone, M., Rainoldi, A., & Boccia, G. (2019). Relevance of evaluating the rate of torque development in ballistic contractions of submaximal amplitude. Physiological measurement, 40(2), 025002.

Boccia, G., Dardanello, D., Brustio, P. R., Tarperi, C., Festa, L., Zoppirolli, C., ... & Rainoldi, A. (2018). Neuromuscular fatigue does not impair the rate of force development in ballistic contractions of submaximal amplitudes. Frontiers in physiology, 9, 1503.

RE: The literatures introduce by the reviewer are not investigated the effect of bilateral contraction on rapid force performance at low and middle force levels. Therefore, it is not accurate to cite those articles here.

Lines 70-71, only using the force data would still be difficult to have a better understanding of BLD.

RE: According to the reviewer’s comment, the sentence of “Clarifying the effect of bilateral contraction on co-regulating rapid force production ability and accuracy will provide an improved understanding of the BLD.” was deleted.

Lines 72-73, what made the author have such hypothesis? Please comment and clarify.

RE: According to the reviewer’s comment, I changed the sentence of hypothesis. (Lines 74-76)

Methods

Participants: how was the sample size (11) determined?

RE: Sample size was determined from participants who were able to participate during the experiment period. 

It was also reported that females and males may have the different bilateral index, especially for the hand muscle (see study: Ye, X., Miller, W. M., Jeon, S., & Carr, J. C. (2019). Sex comparisons of the bilateral deficit in proximal and distal upper body limb muscles. Human movement science, 64, 329-337.). Please justify why you used the mixed sexes for your experiment.

RE: The previous study reported that sex does not influence the magnitude or direction of the bilateral index in the handgrip contraction. Thus, influence of sex on the bilateral index remain unclear. In the present study, I recruited without gender distinction, and did not screening by gender. 

Carr, J. C., Bemben, M. G., Black, C. D., Ye, X., & DeFreitas, J. M. (2020). Bilateral deficit in strength but not rapid force during maximal handgrip contractions. European Journal of Sport Science, 1-8.

Lines 121-124, and lines 155-157, it was suggested that targeting the force levels can slow the rate of force production (Gordon, J., Ghez, C. Trajectory control in targeted force impulses. Exp Brain Res 67, 241–252 (1987). https://doi.org/10.1007/BF00248546). How can the author ensure the RFD in the current study was not influenced by the experimental setup?

RE: The purpose of the present study was to clarify the effect of bilateral contraction on the ability and accuracy of rapid force production in the situation where the submaximal target force levels (30% and 50% MVC). The previous study has only given instructions on either “as fast as possible” or “as accurate as possible”. However, in the present study, wherein participants were instructed to grab a dynamometer with 30% and 50% of MVC as fast and precisely as possible. Thus, it is difficult to simply compare to effect of experimental setup on RFD between previous study and present study. Moreover, in the present study, familiarization session is set up to get used to being able to exert rapid force under the submaximal target force levels.

Line 165, please explain the “32-point window”, how long is this window?

RE: Sampling rate of force signal was 1kHz. Thus, “32-point window” is 32ms.

In addition, when calculating RFD, how was the onset of the force signal determined? I could not find the description.

RE: The peak RFD was calculated from the first derivatives of the force signals. Therefore, the peak RFD is calculated without defining of force onset.

Line 170, when calculating force error, what was the time window used in the calculations?

RE: Force errors were determined as the difference between the target force and peak force in each trial. Thus, when calculating force error, the time window was not used in the calculations. 

Results and Discussion

Overall, the results are hard to follow.

Line 207, Cohen’s d = 0.11, belongs to trivial size.

RE: According to the reviewer’s comment,　I changed to trivial size. (Line 210)

Please change all the “effect size index” to “effect size”. Also, consider using small, medium, large to describe the effect size.

RE: According to the reviewer’s comment,　I changed “effect size index” to “effect size”. Moreover, I changed the notation of effect size to small, medium, large.

For the discussion, limitations of the experiment should be pointed out.

RE: According to the reviewer’s comment, I added limitations of experiment in the discussion section. (Line 392-401)

---

## [Decision Letter · Decision Letter 1]

8 Jan 2021

PONE-D-19-35654R1

Effect of bilateral contraction on the ability and accuracy of rapid force production at submaximal force level

PLOS ONE

Dear Dr. Ohta,

Thank you for submitting your manuscript to PLOS ONE. After careful consideration, we feel that it has merit but does not fully meet PLOS ONE’s publication criteria as it currently stands. Therefore, we invite you to submit a revised version of the manuscript that addresses the points raised during the review process

We look forward to receiving your revised manuscript.

Kind regards,

Riccardo Di Giminiani

Academic Editor

PLOS ONE

Reviewers' comments:

Reviewer's Responses to Questions

**Comments to the Author**

1. If the authors have adequately addressed your comments raised in a previous round of review and you feel that this manuscript is now acceptable for publication, you may indicate that here to bypass the “Comments to the Author” section, enter your conflict of interest statement in the “Confidential to Editor” section, and submit your "Accept" recommendation.

Reviewer #1: All comments have been addressed

Reviewer #2: All comments have been addressed

2. Is the manuscript technically sound, and do the data support the conclusions?

Reviewer #1: Yes

Reviewer #2: Yes

3. Has the statistical analysis been performed appropriately and rigorously? 

Reviewer #1: Yes

Reviewer #2: Yes

4. Have the authors made all data underlying the findings in their manuscript fully available?

Reviewer #1: Yes

Reviewer #2: Yes

5. Is the manuscript presented in an intelligible fashion and written in standard English?

Reviewer #1: Yes

Reviewer #2: Yes

6. Review Comments to the Author

Reviewer #1: The authors have addressed all concerns. The authors should consider using linear mixed models to analyze individual trials in any future work on this topic, rather than averaging trials and using ANOVA as in the current study.

Reviewer #2: Thank you for addressing my previous comments and revising the manuscript accordingly. Most of the comments were well-addressed.

Previously, I asked "how was the sample size (11) determined?", because I wanted to see if the author had run a power analysis. The current sample size might be underpowered. If this is the case, then it should be mentioned in the Limitation.

Previously, I asked "How can the author ensure the RFD in the current study was not influenced by the experimental setup?". I am fine with the response. The author also mentioned "Moreover, in the present study, familiarization session is set up to get used to being able to exert rapid force under the submaximal target force levels." So, please briefly describe what the subjects did during the familiarization.

I saw from Figure 2, that how force error was calculated. What if the actual force was below the target force (the 30% or 50% of the MVC)?

7. PLOS authors have the option to publish the peer review history of their article (what does this mean?). If published, this will include your full peer review and any attached files.

Reviewer #1: No

Reviewer #2: No

---

## [Author Response · Author response to Decision Letter 1]

11 Jan 2021

Response of the Reviewers' comments:

 Thank you very much for your consideration and valuable suggestions for improving my manuscript. I have read the reviewers’ comments very carefully, and have made the following changes according to the comments and suggestions of the reviewers. The revised page numbers and line numbers are indicated following my response to each of the comments. I hope that the revised manuscript satisfactorily addresses the issues raised by the reviewers. My responses are shown in text starting with RE.

Response to Reviewer: 1

Reviewer #1: The authors have addressed all concerns. The authors should consider using linear mixed models to analyze individual trials in any future work on this topic, rather than averaging trials and using ANOVA as in the current study.

RE: Thank you for your advice to the future work. I’d like to use your advice as a reference.

Response to Reviewer: 2

Reviewer #2: Thank you for addressing my previous comments and revising the manuscript accordingly. Most of the comments were well-addressed.

Previously, I asked "how was the sample size (11) determined?", because I wanted to see if the author had run a power analysis. The current sample size might be underpowered. If this is the case, then it should be mentioned in the Limitation.

RE: According to the reviewer’s comment,　as a result of having calculated statistics power, I confirmed that sample size was small. Thus, I added literatures in Limitation section (Line 393-394).

Previously, I asked "How can the author ensure the RFD in the current study was not influenced by the experimental setup?". I am fine with the response. The author also mentioned "Moreover, in the present study, familiarization session is set up to get used to being able to exert rapid force under the submaximal target force levels." So, please briefly describe what the subjects did during the familiarization.

RE: About familiarization section, explanation is listed in line 108-116.

I saw from Figure 2, that how force error was calculated. What if the actual force was below the target force (the 30% or 50% of the MVC)?

RE: When peak force was lower than the target force, the force error showed negative values. This explanation is listed in line 176-177

---

## [Decision Letter · Decision Letter 2]

2 Feb 2021

Effect of bilateral contraction on the ability and accuracy of rapid force production at submaximal force level

PONE-D-19-35654R2

Dear Dr. Ohta,

We’re pleased to inform you that your manuscript has been judged scientifically suitable for publication and will be formally accepted for publication once it meets all outstanding technical requirements.

Kind regards,

Riccardo Di Giminiani

Academic Editor

PLOS ONE

Additional Editor Comments (optional):

Reviewers' comments:

Reviewer's Responses to Questions

**Comments to the Author**

1. If the authors have adequately addressed your comments raised in a previous round of review and you feel that this manuscript is now acceptable for publication, you may indicate that here to bypass the “Comments to the Author” section, enter your conflict of interest statement in the “Confidential to Editor” section, and submit your "Accept" recommendation.

Reviewer #2: All comments have been addressed

2. Is the manuscript technically sound, and do the data support the conclusions?

Reviewer #2: Yes

3. Has the statistical analysis been performed appropriately and rigorously? 

Reviewer #2: Yes

4. Have the authors made all data underlying the findings in their manuscript fully available?

Reviewer #2: Yes

5. Is the manuscript presented in an intelligible fashion and written in standard English?

Reviewer #2: Yes

6. Review Comments to the Author

Reviewer #2: Thank you for the patience. All the comments have been addressed now. I have now further questions or concerns.

7. PLOS authors have the option to publish the peer review history of their article (what does this mean?). If published, this will include your full peer review and any attached files.

Reviewer #2: No

---

## [Editor Report · Acceptance letter]

5 Feb 2021

PONE-D-19-35654R2 

Effect of bilateral contraction on the ability and accuracy of rapid force production at submaximal force level. 

Dear Dr. Ohta:

I'm pleased to inform you that your manuscript has been deemed suitable for publication in PLOS ONE. Congratulations! Your manuscript is now with our production department. 

Kind regards, 

on behalf of

Prof. Riccardo Di Giminiani 

Academic Editor

PLOS ONE